# Trastuzumab Modified Barium Ferrite Magnetic Nanoparticles Labeled with Radium-223: A New Potential Radiobioconjugate for Alpha Radioimmunotherapy

**DOI:** 10.3390/nano10102067

**Published:** 2020-10-20

**Authors:** Weronika Gawęda, Marek Pruszyński, Edyta Cędrowska, Magdalena Rodak, Agnieszka Majkowska-Pilip, Damian Gaweł, Frank Bruchertseifer, Alfred Morgenstern, Aleksander Bilewicz

**Affiliations:** 1Institute of Nuclear Chemistry and Technology, Dorodna 16 Str., 03-195 Warsaw, Poland; w.maliszewska@ichtj.waw.pl (W.G.); m.pruszynski@ichtj.waw.pl (M.P.); e.leszczuk@ichtj.waw.pl (E.C.); m.rodak@ichtj.waw.pl (M.R.); a.majkowska@ichtj.waw.pl (A.M.-P.); 2Faculty of Chemistry, University of Warsaw, Pasteura 1, 02-093 Warsaw, Poland; 3Department of Immunohematology, Centre of Postgraduate Medical Education, Marymoncka 99/103, 01-813 Warsaw, Poland; damian.gawel@cmkp.edu.pl; 4European Commission, Joint Research Centre, Directorate for Nuclear Safety and Security, 76125 Karlsruhe, Germany; Frank.BRUCHERTSEIFER@ec.europa.eu (F.B.); alfred.morgenstern@ec.europa.eu (A.M.)

**Keywords:** alpha radioimmunotherapy, radium-223, barium ferrite

## Abstract

Barium ferrite nanoparticles (BaFeNPs) were investigated as vehicles for ^223^Ra radionuclide in targeted α-therapy. BaFe nanoparticles were labeled using a hydrothermal Ba^2+^ cations replacement by ^223^Ra with yield reaching 61.3 ± 1.8%. Radiolabeled nanoparticles were functionalized with 3-phosphonopropionic acid (CEPA) linker followed by covalent conjugation to trastuzumab (Herceptin^®^). Thermogravimetric analysis and radiometric method with the use of [^131^I]-labeled trastuzumab revealed that on average 19–21 molecules of trastuzumab are attached to the surface of one BaFe–CEPA nanoparticle. The hydrodynamic diameter of BaFe–CEPA–trastuzumab conjugate is 99.9 ± 3.0 nm in water and increases to 218.3 ± 3.7 nm in PBS buffer, and the zeta potential varies from +27.2 ± 0.7 mV in water to −8.8 ± 0.7 in PBS buffer. The [^223^Ra]BaFe–CEPA–trastuzumab radiobioconjugate almost quantitatively retained ^223^Ra (>98%) and about 96% of ^211^Bi and 94% of ^211^Pb over 30 days. The obtained radiobioconjugate exhibited high affinity, cell internalization and cytotoxicity towards the human ovarian adenocarcinoma SKOV-3 cells overexpressing HER2 receptor. Confocal studies indicated that [^223^Ra]BaFe–CEPA–trastuzumab was located in peri-nuclear space. High cytotoxicity of the [^223^Ra]BaFe–CEPA–trastuzumab bioconjugate was confirmed by radiotoxicity studies on SKOV-3 cell monolayers and 3D-spheroids. In addition, the magnetic properties of the radiobioconjugate should allow for its use in guide drug delivery driven by magnetic field gradient.

## 1. Introduction

One of the most important problems in cancer treatment is the therapy of metastatic disease that could efficiently be achieved by internal radiotherapy using alpha-emitting radionuclides. The α particle emitted from a decaying radionuclide is a ^4^He nucleus with mass more than 7000-times heavier than β^−^ particle, and thus it has much shorter range in tissues. Thus, α particles have approximately 500 times greater cytotoxic power than β^−^ particles [1], and only 15 alpha pathways through the cell nucleus are sufficient to induce apoptosis [2]. Additionally, the short range of α particles, 50–100 µm, enables local irradiation of target cells with negligible irradiation of neighboring healthy tissues [3]. Several review articles provide detailed information on the results of these studies [4,5,6]. Recently published reports on the very high therapeutic efficacy of ^225^Ac-labeled PSMA-617 in the treatment of metastatic castration-resistant prostate cancer highlight the clinical potency of alpha radionuclide therapy [7,8].

While hundreds of radionuclides are decayed by the emission of α particle, unfortunately most of these α-emitters have half-lives too short or too long to be applied therapeutically, their production is too expensive and their chemical properties, as in the case of polonium isotopes, do not allow their use in medicine. Actually, the main α-emitters used in targeted therapy are as follows: ^223^Ra, ^211^At, ^225^Ac, ^212^Pb, ^226,227^Th and ^212,213^Bi.

^223^Ra is the first alpha emitter applied in radionuclide therapy. ^223^Ra as radium dichloride ([^223^Ra]RaCl_2_) is the first α-emitting drug which has a marketing authorization from the United States Food and Drug Administration (FDA) for the treatment of bone metastases [9]. However, Ra^2+^, similar to other cations of the elements of the alkali earth group of the periodic table, forms only weak complexes; therefore, labeling of the biological vector with ^223^Ra is challenging. Up to now, the stability constant of the Ra with 1,4,7,10-Tetraazacyclododecane-1,4,7,10-tetraacetic acid (DOTA)complex has not been determined, but, based on the value for Ba^2+^ (log*K* = 12.6) [10], it can be estimated that the log*K* for Ra^2+^ is around 12, whereas for Ac^3+^ it is 10 orders of magnitude higher.

Performed studies on complexation of radium cations by aminocarboxylic acids, cryptands and calixarenes were unsuccessful. Henriksen et al. [11] studied the most promising ligands for ^223^Ra^2+^ complexation: cyclic polydentate ligands DOTA, Kryptofix 2.2.2, calix[4]-tetraacetic acid and the linear chelator diethylenetriaminepentaacetic acid (DTPA). Based on the determined extraction constants, calix[4]-tetraacetic seemed to be the most promising of the compounds studied. Recently, Mamat et al. [12] evaluated polyoxopalladates for incorporation of radium radionuclides in the center of the complex. In 2020, a short report was published on the possibility of application a macrocycle ligand having a large complexation ring (MACROPA) for ^223^Ra complexation [13]. Initial results are promising. In the biodistribution studies, ^223^Ra-MACROPA displayed a striking contrast in bone uptake with respect to ^223^RaCl_2_ administration, revealing only 1.6 ± 0.3% IA/g versus 22 ± 1% IA/g in femoral uptake after 24 h p.i.

In the case of radionuclides which are the parent radionuclides for the decay chains such as ^225^Ac, ^223^Ra and ^227^Th, the designed ligand should form strong complexes with the mother and also with the daughter radionuclides. ^225^Ac radionuclide first decays to ^221^Fr (alkali metal) and escapes from radiobioconjugate. Similarly, in the case of ^227^Th and ^223^Ra radionuclides, the first decay products, ^223^Ra and ^219^Rn, respectively, liberate themselves from ^227^Th and ^223^Ra. This problem is less relevant for ^223^Ra decay series because 75% of its alpha energy are emitted within a few seconds (t_1/2_ = 4 s) after the decay of ^223^Ra [14]. Therefore, to expand the application of ^223^Ra beyond bone metastases therapy, instead of bifunctional ligands, series of nanoparticles were studied for immobilization of ^223^Ra and its binding to a biomolecule. In the first studies, liposomes were used to immobilize ^223^Ra. ^223^Ra was immobilized in liposomes with a high efficiency and retained ^223^Ra for long time in serum [14]. In subsequent studies using liposomal doxorubicin Caelyx^®^/Doxil^®^ drug, Larsen et al. [15] synthesized radiobioconjugates containing ^223^Ra inside liposomes functionalized by two targeting biomolecules: folic acid and monoclonal antibody fragment F(ab′)_2_(IgG1). In the biodistribution studies performed, they found that adsorption on the bones increased with time because ^223^Ra@liposome was metabolized. As a result of this process, cationic ^223^Ra^2+^ was liberated, which quickly adsorbed onto the bone surface. Unfortunately, these promising studies on the use of liposomes to immobilize ^223^Ra were not continued.

The next studies focused on the possibilities of application inorganic nanoconstruct to immobilize ^223^Ra on the nanoparticles surface or their incorporation into the nanostructure. Several studies have used the similarity between the Ba^2+^ and Ra^2+^ to incorporate ^223^Ra into the crystal structure of the insoluble barium compounds. The ionic radii of the Ba^2+^ and Ra^2+^ cations are almost identical, 142 and 148 pm, respectively [16]. Therefore, it is easy to replace the cation of Ba^2+^ by Ra^2+^. Reissig et al. [17,18] proposed application of BaSO_4_ nanoparticles to stably bind ^224^Ra to targeting vectors. Unfortunately, during synthesis of [^224^Ra]BaSO_4_ nanoparticles, only 20% of ^224^Ra radionuclide was immobilized into the BaSO_4_ nanoparticles, whereas 80% of the activity remained free in the solution.

In a series of publications, Kozempel et al. [19,20,21,22] tested hydroxyapatite (HAP) nanoparticles as a transporter for ^223^Ra radionuclide. They studied two methods for binding of ^223^Ra to HAP nanoparticles. In the first method, they used the surface ^223^Ra adsorption on synthesized HAP nanoparticles, while the second was based on incorporation of ^223^Ra into HAP structure during their synthesis processes. The labeling yields in both methods were 60–99% and leaching of ^223^Ra was negligible, although the internal method gave slightly better labeling. A surprisingly small release of ^223^Ra and decay products ^211^Bi and ^211^Pb was detected over the stability test, probably due to resorption of liberated ^223^Ra and ^211^Pb on the surface of HAP nanoparticles. This situation was also observed in the case of nanozeolites [23] where resorption causes large retention of ^223^Ra and ^223^Ra decay products.

A new approach for ^223^Ra conjugation to biomolecules was proposed by Piotrowska et al. [23,24] by application NaA zeolites nanoparticles as ^223^Ra transporter. In nanozeolite bioconjugates, ^223^Ra labeling occurs by ion exchange process after synthesis of the ready radiopharmaceutical, while, in the other proposed methods, nanoparticles are first labeled and then conjugated to biomolecule. NaA nanozeolites were conjugated to 5–11 fragment of substance P, a peptide exhibiting affinity to NK1 receptor on glioblastoma cells and anti-PSMA D2B antibody [24,25]. The obtained ^223^RaA–silane–PEG–SP(5–11) bioconjugate was stable, keeping >98% of ^223^Ra and >94% of the decay radionuclides.

After developing the method for immobilization of ^225^Ac inside lanthanides phosphate and vanadate nanoparticles [26,27,28,29,30], a team from the University of Missouri performed studies on the application of LnPO_4_ nanoparticles to incorporate ^223^Ra and ^225^Ra [31]. The ^223^Ra labeled LaPO_4_ retained ~88% of the ^223^Ra for 35 days, while, after covering of the core by two LaPO_4_ layers, it decreased the release of ^223^Ra and its decay products, ^211^Pb, to 0.1%.

In the present study, as a transporter for ^223^Ra, we proposed to use barium ferrite BaFe_12_O_19_ (BaFe) nanoparticles where Ba^2+^ cations can be replaced by Ra^2+^ in their magnetic core. Hexagonal ferrites such as BaFe_12_O_19_ are known for their ferromagnetic properties and are used in magnetic recording materials [32]. Similar to magnetite nanoparticles (SPIONs), BaFe nanoparticles can also be used in nanomedicine, especially in targeted drug delivery, MRI bioimaging diagnostics and magnetic hyperthermia [33,34]. It can be assumed that ^223^Ra-doped BaFe nanoparticles should be promising candidates for multimodal drug combining localized magnetic hyperthermia with internal α-therapy. The synergy of ionizing radiation and magnetic hyperthermia in the destruction of tumor cells has been confirmed many times [35]. Combining these therapeutic methods, one expects a reduction in the required delivered radiation dose needed, or to increase the effect of radiation on the cellular hypoxia found in tumors by enhancing the energy deposition [36]. It is also worth emphasizing that hyperthermia is complementary with radiotherapy because elevated temperatures mainly act on cells in S phase, when the ionizing radiation in the phase G2 and M in the cell cycle. In previous studies, combinations of local hyperthermia with external sources of radiation, such as γ rays or electron beams, have been applied [34]. In our solution, the emissions of corpuscular radiation and heat occurs from the same nanoparticle and thus from within the tumor.

## 2. Materials and Methods

### 2.1. Chemical Reagents

The used chemicals were obtained from Sigma-Aldrich (St. Louis, MO, USA without further purification: iron(III) chloride hexahydrate (ACS reagent, 97%), 3-phosphonopropionic acid (CEPA, 94%) and human serum (stored at −22 °C), 2-(*N*-morpholino)ethanesulfonic acid (MES, ≥99%). Iodogen (1,3,4,6-tetrachloro-3α,6α-diphenyl-glycoluril),*N*-(3-dimethylaminopropyl)-*N*′-ethylcarbo-diimide hydrochloride (EDC, ≥99%) and *N*-hydroxysulfosuccinimide sodium salt (sulfo-NHS, ≥98%) were obtained from Thermo Scientific (Rockford, IL, USA). Monoclonal antibody, trastuzumab, was separated from the commercial product (Herceptin^®^, Roche Pharmaceuticals, Basel, Switzerland) on Vivaspin 500 (Sartorius, Stonehouse, UK) with a 50-kDa cutoff membrane. Aqueous solutions were obtained from ultrapure deionized water prepared using the Milli-Q filtering system (Merck, Darmstadt, Germany).

### 2.2. Radionuclides

^223^Ra was obtained by chemical isolation from ^227^Ac source obtained from the European Commission Joint Research Centre (Karlsruhe, Germany) in the amount of 6 MBq. No-carrier-added sodium [^131^I]iodide was supplied from the Radioisotope Centre POLATOM (Świerk, Poland).

### 2.3. Cell Lines

SKOV-3 (HER2-positive) and MDA-MB-231 (HER2-negative) cell lines were obtained from the American Type Culture Collection (ATCC, Rockville, MD, USA) and cultured in McCoy’s 5A and DMEM mediums, respectively. The cell culture mediums were enriched with 10% heat-inactivated fetal bovine serum, L-glutamine, streptomycin (100 μg/mL) and penicillin (100 IU/mL). Cells were cultured in a humidified atmosphere with 5% CO_2_ at 37 °C according to ATCC protocol. Prior to their in vitro use, cells were detached using trypsin–EDTA (0.25%). All reagents for cell growth were purchased from Biological Industries (Beth Haemek, Israel).

### 2.4. Instrumentation

The morphology (size and shape) of synthesized BaFe_12_O_19_ nanoparticles (BaFeNPs) and obtained conjugates was evaluated with transmission electron microscopy (TEM, Zeiss Libra 120 Plus, Stuttgart, Germany). The zeta potential (ζ) and hydrodynamic diameter of BaFeNPs, before and after every surface modification (first with 3-phosphonopropionic acid (CEPA), and then with trastuzumab), were determined with dynamic light scattering (Zetasizer Nano ZS DLS, Malvern, UK). The modification of the BaFeNP surface was also evaluated with thermogravimetric analysis using SDT-Q600 Simultaneous TGA instrument with TA Universal Analysis software (TA Instruments, New Castle, DE, USA). The magnetic properties of samples were studied using QD vibrating sample magnetometer VSM (NanoMagnetics Instruments, Oxford, UK) over the magnetic field from −2.0 to +2.0 T at temperatures of 100–300 K with an accuracy of ca. 0.01 K. The powder X-ray diffraction patterns were recorded with an X-ray diffractometer (PXRD, Rigaku SmartLab 3 kW, Tokyo, Japan) operating with Dtex detector, Cu Kα radiation, line λ = 1.541837 Å and a scan rate of 1° per minute in 0.02° steps covering the 2° angle range 2theta from 10° to 80°. Measurements were performed at room temperature (RT).

The concentration of purified trastuzumab used in conjugation synthesis was determined spectrometrically using UV-Vis Evolution 600 spectrophotometer (Thermo Fisher Scientific, Madison, WI, USA) following the method proposed by Maeda et al. [37] with a molar extinction coefficient (ε) of 225,000 M^−1^ cm^−1^ at 280 nm. The radioactivity measurement was performed by γ-spectrometry on a Coaxial High Purity Germanium (HPGe) detector (GX 1080) connected to a multichannel analyzer DSA-1000 (Canberra, Meriden, CT, USA). The following gamma lines were applied: 269.4 keV for ^223^Ra, 832.0 keV for ^211^Pb and 351.0 keV for ^211^Bi. In the cell studies, the radioactivity was measured using a Wizard 2^®^ automatic gamma counter 2480 (Perkin Elmer, MA, USA).

Raw data and statistical analyses in all in vitro experiments were carried using GraphPad Prism 7.0 software (GraphPad Software Inc., San Diego, CA, USA). Data obtained from cell study experiments were evaluated with Student’s *t*-distribution test. When the *p*-value was less than <0.05, differences were considered statistically significant.

### 2.5. Synthesis of Barium Ferrite Nanoparticles Labeled with ^223^Ra

Barium ferrite magnetic nanoparticles were synthesized using the modified autoclave method proposed by Drofenik et al. [38]. The scheme of the synthesis is presented in Figure 1.

The synthesis was carried out in a 25 mL stainless steel high-pressure reactor with PTFE insert (Berghof Products+Instruments GmbH, Eningen unter Achalm, Germany). Aqueous solutions of 0.0126 M barium and iron chlorides were prepared by dissolving their salts in distilled water. Next, 2.777 mL of iron chloride and 0.347 mL of barium chloride were added to the PTFE vessel, giving the molar ratio of Fe^3+^:Ba^2+^ equal to 8:1. Finally, 0.875 mL of 2.6 M sodium hydroxide were added to the solution, and the PTFE vessel with reaction mixture was placed in the autoclave which was then heated to 210 °C. After reaching the required temperature, the synthesis was carried out for 5 h with continuous stirring. The obtained product was separated from the reaction mixture by a strong solid magnet and washed several times with distilled water and 1 mM hydrochloric acid. Finally, the prepared nanoparticles were dispersed in distilled water and used for further studies.

The ^223^Ra-labeled barium ferrite nanoparticles ([^223^Ra]BaFeNPs) were synthesized in a similar way as the non-radioactive ones. Synthesis was performed in an autoclave using the same amounts of barium and iron chlorides as described above. Then, before adding sodium hydroxide, 0.1–0.5 MBq of ^223^Ra as a chloride salt was added to the solution. The obtained radioactive nanoparticles were separated from reaction mixture by a strong solid magnet and washed several times with distilled water and with 1 mM hydrochloric acid. The percentage of ^223^Ra incorporated into barium ferrite crystalline structure was calculated from the ratio of radioactivity retained inside nanoparticles to radioactivity of ^223^Ra initially added to the solution.

### 2.6. Synthesis of [^223^Ra]BaFe–CEPA–Trastuzumab

The bioconjugation of BaFeNPs and [^223^Ra]BaFeNPs with trastuzumab monoclonal antibody was carried out in two steps according to our previous protocol [39]. Firstly, 1 mL of 30 mg/mL 3-phosphonopropionic acid (CEPA) was mixed with 4.5–5.0 mL of 0.1 M sodium hydroxide until the pH reached 7. Next, 1 mL of 1 mg/mL solution of BaFeNPs or [^223^Ra]BaFeNPs was added drop-wise to the solution of CEPA with sodium hydroxide and stirred at room temperature (RT). The suspension was sonicated for 25 min in an ultrasonic bath. Finally, the nanoparticles were separated from reaction mixture by a strong magnet and washed several times with distilled water. Nanoparticles with modified by CEPA surface were dispersed in 500 μL of water and mixed with 500 μL of 0.5 M MES buffer, pH 6.1, containing 0.297 μmol of sulfo-NHS and 0.297 μmol of EDC. The prepared solution was stirred for 4 h at RT. After activation of carboxylic groups in CEPA to NHS-ester, the nanoparticles were isolated from the synthesis solution on the solid magnet, washed a few times with 10 mM PBS and resuspended in 1 mL of 10 mM PBS, pH 7.4. Then, 250 μg (1.7 nmoles) of trastuzumab were added to the suspended nanoparticles. The synthesis of bioconjugate with trastuzumab was performed overnight. The conjugation product was isolated from the solution using the magnet and washed with distilled water.

### 2.7. Stability Studies of [^223^Ra]BaFeNPs and [^223^Ra]BaFe–CEPA–Trastuzumab

Synthesized [^223^Ra]BaFeNPs and [^223^Ra]BaFe–CEPA–trastuzumab radiobioconjugate were tested for leaching of the ^223^Ra and its decay products using the procedure described by Piotrowska et al. [24]. Briefly, a portion of suspended radiolabeled nanoparticles were introduced into the Eppendorf tubes with 1 mL of the following solutions: 0.9% NaCl, 1 mM PBS, human serum (HS) and McCoy’s 5A cell culture medium. In addition, a blank test with the serum and radionuclide solution was performed to estimate the percentage of possible binding of free radionuclides cations ^223^Ra^2+^, ^211^Pb^2+^ and ^211^Bi^3+^ to proteins present in the serum and on the surface of the vials. The stability studies were carried out in static conditions by mixing in the rotating stirrer up to 30 days. Each day, a sample was centrifuged and the obtained supernatant was measured for ^223^Ra and its decay products. The activity of ^223^Ra and its decay products ^211^Pb and ^211^Bi released from nanoparticles and radiobioconjugate was determined by HPGe γ-spectrometry. The percentage of retention was determined as a ratio of the radioactivity in the supernatant to the radioactivity of the whole probe.

### 2.8. Quantification of the Number of Trastuzumab Molecules per One Barium Ferrite Nanoparticle

The number of trastuzumab molecules conjugated to each barium ferrite nanoparticle was determined by two methods: thermogravimetric analysis (TGA) and using radioiodinated trastuzumab. The obtained compounds, namely BaFeNPs, BaFe–CEPA and BaFe–CEPA–trastuzumab bioconjugate, were dried at RT. After drying, 3 mg of each samples were placed in a TGA furnace and heated at the rate of 10 °C/min. The mass changes in the synthesized nanoparticle samples were determined in the temperature range 20–800 °C under inert gas (nitrogen) flow. The flow of the gas was established at the rate of 100 mL/min.

The second procedure was based on attaching [^131^I]-labeled trastuzumab to nanoparticles by using the procedure described earlier with some modifications [39]. Briefly, trastuzumab (1 mg) in 200 µL of 0.01 M PBS was labeled with radionuclide ^131^I (10–15 MBq) by using tubes coated with 10 µg of dried Iodogen. After 10 min of reaction, the obtained [^131^I]-trastuzumab was purified on PD-10 columns filled with Sephadex G-25 (GE Healthcare Life Sciences, Piscataway, NJ, USA). In the second step, 250 μg of [^131^I]-trastuzumab were added to cold barium ferrite nanoparticles coated by 3-phosphonopropionic with NHS-activated groups and mixed overnight, as described above. After 24 h, nanoparticles conjugated with [^131^I]-trastuzumab were isolated from the solution using a solid magnet, washed several times with water and resuspended in distilled water. To calculate the number of [^131^I]-trastuzumab attached to each BaFe–CEPA nanoparticle, the conjugated moles of [^131^I]-trastuzumab were divided by the moles of used nanoparticles.

### 2.9. Binding Specificity Assay

The affinity of [^223^Ra]BaFe–CEPA and [^223^Ra]BaFe–CEPA–trastuzumab radiobioconjugates to bind HER2 receptors was tested on SKOV-3 (HER2+, HER2-overexpressing cell lines) and for comparison on the HER-2-negative MDA-MB-231 cells. Briefly, 8 × 10^4^ cells per well were seeded into 24-well plates and cultured at 37 °C overnight. After 24 h, the cells were washed twice with PBS, and [^223^Ra]BaFe–CEPA or [^223^Ra]BaFe–CEPA–trastuzumab radiobioconjugate suspended in medium was added to each well and incubated at 4 °C for 2 h. Next, the supernatant was removed, and cells were washed twice with cold PBS. Finally, the cells were lysed twice with 1 M sodium hydroxide. Non-specific binding was determined by co-incubation with 100-fold excess of cold trastuzumab (blocking experiment). Collected fractions of [^223^Ra]BaFe–CEPA–trastuzumab radiobioconjugate unbound and bound to the receptors were measured on a Wizard 2^®^ automatic gamma-counter.

### 2.10. Internalization Studies

The internalization studies were performed for [^223^Ra]BaFe–CEPA–trastuzumab radiobioconjugate on SKOV-3 cell line. Cells were seeded into 6-well plates (TPP, Switzerland) at the density of 5 × 10^5^ cells per well and cultured for 24 h. Next, cells were incubated with [^223^Ra]BaFe–CEPA–trastuzumab radiobioconjugate at the concentration of 0.25 μg/mL for 1 h at 4 °C. After 1 h, the cell culture medium was removed and collected, and the cells were washed twice with PBS and incubated with fresh medium at 37 °C for another 1, 6 and 24 h. After each time point, culturing medium was collected, each well was washed twice with glycine pH 2.8 (for stripping the unbound radiobioconjugate from the cells membrane) and then with 1 M sodium hydroxide for collecting the cells (internalized fraction). Each collected fraction was measured on a Wizard 2^®^ automatic gamma-counter. The results are expressed as a percentage of the initially bound radioactivity.

### 2.11. Confocal Imaging

SKOV-3 cells (2 × 10^5^ per well) were seeded in 6-well plates covered with sterile glass coverslips (ϕ12 mm/#1.5; Thermo Scientific, San Jose, CA, USA). The next day, the medium was removed and the following compounds suspended in cell culture medium were added to wells: BaFeNPs (~3.7 × 10^12^ nanoparticles/well), BaFe–CEPA (~3.7 × 10^12^ nanoparticles/well) and BaFe–CEPA–trastuzumab (~3.7 × 10^12^ nanoparticles/well). Cells cultured in the presence of medium only were used as a control. After 24 h of incubation with various compounds, the coverslips were moved to 24-well plates, washed, fixed with 4% PFA, permeabilized and blocked as described previously [40]. Further, cells were stained with 4′,6-diamino-2-phenylindole dihydrochloride (DAPI; Sigma-Aldrich). Finally, after washing with deionized water, coverslips were mounted using Fluorescence Mounting Medium (Dako, Carpinteria, CA, USA). Microscope examination was performed using the confocal microscope LSM800 with plan-apochromat 63/1.4 oil DIC M27 lens (Zeiss, Jena, Germany). Fluorescence emission of FITC-labeled trastuzumab and DAPI were measured at 488 and 408 nm, respectively. Transmitted light images in the bright field for BaFeNPs or BaFe–CEPA–trastuzumab were acquired using a transmitted light detector (T-PMT).

### 2.12. In Vitro Cytotoxicity Assay

To assess cell metabolic activity, the MTS assay was used. The cytotoxicity of the synthesized BaFeNPs, BaFe–CEPA–trastuzumab bioconjugate, [^223^Ra]BaFe–CEPA–trastuzumab radiobioconjugate, [^223^Ra]BaFe–CEPA nanoparticles and ^223^Ra radionuclide was evaluated on the SKOV-3 cell line. Cells were seeded in 96-well microplates (TPP, Switzerland) at a density of 2 × 10^3^ cells per well. Afterwards, cells were washed with cold PBS and treated (in triplicates) with increasing concentrations of the radiocompounds (0.1–50.0 kBq/mL) and non-radioactive barium ferrite (0.78–400.00 µg/mL) suspended in cell culture medium at a volume of 100 µL per well. The SKOV-3 cells were incubated with compounds for 2 h at 37 °C. Afterwards, cells were washed twice with PBS and subsequently incubated in the presence of fresh cell culture medium at 37 °C for 24 and 48 h. After each time of incubation, 20 μL of CellTiter 96^®^ AQueous One Solution Reagent (Promega, Mannheim, Germany) were added to each well and incubated for another 2 h at 37 °C. Next, the absorbance of the samples was measured at 490 nm using the Apollo 11LB913 microplate reader (Berthold, Bad Wildbad, Germany). The results are presented as a percentage of cell viability in comparison to the control represented by cells cultured in medium only. The IC_50_ in kBq/mL was calculated by analyzing viability curves with GraphPad Prism (7.0).

### 2.13. Spheroids

The formation of spheroids was initiated by seeding SKOV-3 cells into a 96-well plate with the ultra-low attachment surface (Corning, NY, USA). Spheroids were grown to the size of 250 μm (5 × 10^3^ cells per well seeded). Next, they were treated with 0.78, 3.13, 12.50 and 50.00 kBq/mL of ^223^Ra or [^223^Ra]BaFe–CEPA–trastuzumab. Spheroids were incubated with the radiocompounds for 2 h, washed twice with 100 μL of cold PBS and suspended in fresh medium, which was then replaced every day. The growth of individual spheroids was measured for up to 12 days after treatment. To determine the diameter of spheroids, Primovert microscope with Axiocam 105 color (Zeiss, Jena, Germany) was applied. Measurements were performed with ZEN 2.3 lite software (Zeiss, Jena, Germany), and ImageJ (University of Wisconsin-Madison, Madison, WI, USA) software was used for processing of obtained spheroids images.

## 3. Results and Discussion

### 3.1. Synthesis and Characterization of Barium Ferrite Nanoparticles

Several synthetic methods have been developed for synthesis of barium hexaferrites [41,42,43,44,45,46,47]. However, all these procedures require two basic operations: mixing of initial components mechanically and a subsequent calcination of the obtained mixture with the temperature usually ranging 600–1400 °C. Due to the annealing at high temperatures, the size of barium ferrite grain is usually larger than 100 nm, which limits the possibilities of obtaining ultrafine particles for medical application. Since the main objective of our study was the incorporation of ^223^Ra in small BaFeNPs, we decided to use hydrothermal synthesis (without the calcination step), which allows obtaining BaFeNPs with sizes <50 nm, although with a lower magnetization saturation [44].

The obtained BaFeNPs were characterized for morphology, crystallinity and magnetization. The TEM image reveals that the sample consisted of nearly spherical nanoparticles with diameter about 14–30 nm with mean value of 21.6 ± 3.6 nm (Figure 2). The XRD pattern of the sample obtained at 210 °C is shown in Figure 3. The product was identified as hexagonal BaFe_12_O_19_ with some iron oxide impurities. The impurity phases identified were Fe_2_O_3_ and its hydrate, which could be easily eliminated by washing the product with diluted HCl solution. The broadening of the peaks is evident, indicating significant decrease of the crystallite size. The comparison of the XRD results for the nanoparticles synthesized by us with XRD data of nanoparticles calcined at 800 °C [38,41] indicates a much lower degree of crystallinity of non-calcinated BaFeNPs.

The hysteresis loop of the sample prepared at 210 °C is shown in Figure 4. The saturation magnetization at room temperature (43.36 kA/m) is much higher than the value obtained by Che et al. [47] (5.81 kA/m) for the sample synthesized at 140 °C, but it is significantly lower than for the BaFe bulk material and the nanoparticles calcined at 580 °C (353 kA/m [41]) or superparamagnetic magnetite nanoparticles (446 kA/m [34]). The weak magnetic properties of hydrothermally obtained BaFeNPs are associated with its low crystallinity and, as suggested by Che et al. [47], the high concentration of oxygen vacancies and low Fe^3+^–O^2−^–Fe^3+^ exchange interaction. As mentioned above, our attempts to calcinate BaFeNPs at temperatures >500 °C caused aggregation of the product, which did not allow its further dispersion into nanoparticles.

The low saturation magnetization of hydrothermally synthesized BaFeNPs significantly reduce their applicability in magnetic hyperthermia and MRI imaging, where high saturation magnetization of the samples is needed [48]. However, saturation magnetization of 43.36 kA/m may be sufficient for magnet-guided delivery of ^223^Ra immobilized in BaFeNPs under influence of external magnetic field.

Since ^223^Ra does not undergo strong surface adsorption on hydroxyl groups of magnetite [49] or BaFeNPs, we decided to use intrinsic incorporation of ^223^Ra into the crystalline structure of BaFeNPs. Intrinsic ^223^Ra labeling was previously tested for nanoparticles containing elements chemically similar to Ra, e.g., Ca in hydroxyapatite NPs [19,22] and Ba in BaSO_4_ [18]. This way of nanoparticles labeling provides higher stability than surface adsorption. A schematic reaction running during the synthesis of barium ferrite doped with ^223^Ra^2+^ should go as follows:x ^223^Ra^2+^ + (1 − x) Ba^2+^ + 12 Fe^3+^ + 38 OH^−^ → ^223^Ra_x_Ba_(1−x)_Fe_12_O_19_↓ + 19 H_2_O(1)

Because ^223^Ra is in non-carrier-added form, in the case of samples with specific activity of 1 MBq/mg, the x value in Equation (1) is only 2 × 10^−10^. In the used ^223^Ra radioactivity range 0.1–0.5 MBq/mg of BaFeNPs, the ^223^Ra^2+^ labeling yield reached 61.3 ± 1.8%.

### 3.2. Functionalization of BaFeNPs

The bioconjugation of [^223^Ra]BaFeNPs with trastuzumab monoclonal antibody was performed in two steps. First, a 3-phosphonopropionic acid (CEPA) linker was attached to the BaFeNPs hydroxyl surface groups. Second, the activation of carboxylic group on the CEPA linker to NHS-ester was performed followed by simultaneous coupling to trastuzumab molecule. Each step of surface modification was verified with dynamic light scattering (DLS) analysis. As presented in Table 1 (measured in water) and Table 2 (measured in 1 mM PBS), the hydrodynamic diameters are larger in comparison to the results obtained by TEM analysis. This fact is related to the presence of the solvation layer of water as well as the possible aggregation of bare nanoparticles.

The hydrodynamic diameter of BaFe–CEPA is significantly smaller than that for the bare barium ferrite nanoparticles because being connected to the surface of 3-phosphonopropionic acid molecules containing free carboxyl group stabilizes the nanoparticles and prevents their aggregation. Successful surface modification with 3-phosphonopropionic acid was also confirmed by the change in the zeta potential values. The attachment of CEPA causes a change of zeta potential from positive (protonated surface –OH groups of BaFeNPs) to negative due to the presence of –COO^−^ groups on the surface of BaFe–CEPA nanoparticles. After conjugation of trastuzumab, we observed the change of zeta potential to positive value in water and increase of the zeta potential in 1 mM PBS as a result of the presence of protonated amine groups on the BaFe–CEPA–trastuzumab. Therefore, a change in zeta potential and increase in hydrodynamic diameter confirmed the successful binding of trastuzumab to the surface of barium ferrite nanoparticles.

In addition, thermogravimetric analysis (TGA) confirmed the presence of trastuzumab on the BaFeNPs surface (Figure 5). All thermograms present a slight loss of mass in the range of 20–100 °C, which is related to the desorption of water. The difference of weight loss between BaFeNP and BaFeNP–CEPA samples (2.8%) is related to the release of 3-phosphonopropionic acid molecules present on the barium ferrite surface. In the case of the BaFe–CEPA–trastuzumab bioconjugate, the observed difference in weight loss (5.0%) is related to the release and thermal decomposition of trastuzumab molecules. Based on the TGA results, the number of bound trastuzumab molecules per one barium ferrite nanoparticle was measured. The calculations were carried out assuming that each nanoparticle is spherical and has a diameter of 21.6 nm (measured by TEM) and the BaFe density is 5.28 g·cm^−3^. The obtained results indicate that approximately 19 trastuzumab molecules were coupled to one BaFeNP.

This result was confirmed by radiometric method using radioactive [^131^I]-trastuzumab. Conjugation efficiency was estimated by measuring the ratio of radioactivity bound to nanoparticles to the radioactivity of the initially added [^131^I]-trastuzumab. The number of trastuzumab molecules conjugated to one BaFeNP estimated by this method was 21, which is in good agreement with the results measured by TGA method.

### 3.3. Stability Studies

The stability of synthesized [^223^Ra]BaFeNPs and [^223^Ra]BaFe–CEPA–trastuzumab bioconjugates was tested in physiological saline (0.9% NaCl) and human serum for 30 days (Figure 6). In the control experiment with ^223^Ra, no adsorption of ^223^Ra, ^211^Pb and ^211^Bi was observed on serum proteins or on the surface of the vials. The synthesized [^223^Ra]BaFeNPs retained >97% of ^223^Ra and >95% of ^211^Bi over 30 days in 0.9% NaCl and human serum. Slightly lower retention (88%) was observed in saline solution for ^211^Pb. The [^223^Ra]BaFe–CEPA–trastuzumab bioconjugate retained >95% of ^223^Ra in 0.9% NaCl over 30 days, but about 15% of ^211^Bi and 27% of ^211^Pb was released. In human serum, the radiobioconjugate is more stable, retaining >98% of ^223^Ra, about 96% of ^211^Bi and 94% of ^211^Pb over 30 days.

The ^223^Ra retention inside BaFeNPs was higher than would be expected considering their size. The recoil distance of the first decay product ^219^Rn (t_1/2_ = 3.96 s) in the materials with density similar to BaFe (5.28 g/cm^3^) is about 100 nm [50]; therefore, we can expect that more than 80% of ^219^Rn atoms would be released from 20-nm nanoparticles. However, the released ^219^Rn immediately decays to another two α-emitters, ^211^Pb (t_1/2_ = 36 min) and ^211^Bi (t_1/2_ = 2.17 min), and the formed ^211^Pb^2+^ and ^211^Bi^3+^ cations can be readsorbed on the hydroxyl groups of the BaFeNP surface. Due to the higher oxidation state of Bi^3+^, its readsorption is more effective than Pb^2+^. The ion exchange properties of BaFeNPs are described in a number of publications [51,52]. As in the case of iron oxides in solutions with pH > 5, an effective exchange of proton in hydroxyl groups by multivalent cations takes place. A similar reabsorption process of ^211^Pb and ^211^Bi was also observed in ^223^Ra-labeled hydroxyapatites [22] and nanozeolites [23,24]. Although, according to Holzwart et al. [53], the results obtained in batch measurements, in which the ^223^Ra labeled nanoparticles are equilibrated with biological liquids, cannot be transferred to living organism, where blood flow rapidly dislocates the decay products of ^223^Ra from the surface of the nanoparticles and reduces the possible readsorption, considering the fast and high internalization properties of [^223^Ra]BaFe–CEPA–trastuzumab bioconjugate (see Section 3.5), it can be expect that after internalization in targeted cells the readsorption process will play an essential role in preventing the escape of ^211^Pb and ^211^Bi from cancerous cells [54].

### 3.4. Specificity of Binding

The affinity of [^223^Ra]BaFe–CEPA and [^223^Ra]BaFe–CEPA–trastuzumab radiobioconjugates to HER2 receptors was examined on SKOV-3 (HER2+) and MDA-MB-231 (HER2-) cells. Tested [^223^Ra]BaFe–CEPA and [^223^Ra]BaFe–CEPA–trastuzumab radiobioconjugate were incubated in the presence (blocked) or absence of a 100-fold molar excess of non-labeled trastuzumab. The results are presented in Figure 7.

In the case of [^223^Ra]BaFe–CEPA nanoparticles, as expected, there is no significant difference between the results obtained with or without blocking, on both SKOV-3 (*p* > 0.54) and MDA-MB-231 (*p* > 0.22) cell lines. For the [^223^Ra]BaFe–CEPA–trastuzumab radiobioconjugate, there is no significant difference observed for MDA-MB-231 between non-blocked and blocked cells with 100-fold molar excess of trastuzumab (*p* > 0.34); the radioactivity bound to the cells is 12.84 ± 0.31% for non-blocked and 13.22 ± 0.24% for cells with blocked HER2 receptors. However, when it comes to the SKOV-3 cell line, there is a significant difference (~3.58%) observed between non-blocked and blocked cells (*p* < 0.02); the radioactivity bound to cells is equal to 19.16 ± 0.89% and 15.58 ± 0.19%, respectively. In contrast to SKOV-3, for the MDA-MB-231 cell line, the lack of significant difference in binding between [^223^Ra]BaFe–CEPA and [^223^Ra]BaFe–CEPA–trastuzumab with and without blocking additionally confirms the specificity of binding for our radiobioconjugate. The results obtained for our radiobioconjugate (3.58%) are similar and comparable to those with anit-HER2 ^225^Ac-DOTA-2Rs15d-Nanobody conjugate (5.0%) published by Pruszyński et al. [55]. Unfortunately, for our BaFeNPs and BaFeNP conjugates with trastuzumab, a high background derived from physical sorption of nanoparticles on cell surface during the long-term incubation process was observed in trastuzumab blocked and unblocked cells. It may be associated with sedimentation of tested compounds, as the experiment was carried out in static conditions. To minimize this effect, incubation in our experiments lasted only 2 h. In future experiments, we will try to reduce this effect by using the cell culture under flow instrument.

### 3.5. Internalization Studies

The internalization of [^223^Ra]BaFe–CEPA–trastuzumab radiobioconjugate in SKOV-3 cells was tested at three incubation time points at 37 °C. Due to the lack of specific binding, we did not conduct internalization studies on MDA-MB-231 cells. The obtained results are presented in Figure 8. The quick and high internalization properties of trastuzumab are well-known [56]. In BT-474 cell line (HER2-positive), intracellular activity varies from ~85% at 1 h to ~62% at 24 h [56]. Therefore, it was not surprising that internalization of our radiobioconjugate was very high and equal to 90.60 ± 2.97% already at 1 h, rising to 94.50 ± 0.20% at 6 h of incubation and followed by a slight decrease to 90.00 ± 1.70% at 24 h. These results clearly indicate that ^223^Ra-labeled barium ferrite nanoparticles coupled with trastuzumab were quickly internalized and well retained inside the cells, as the amount of ^223^Ra radioactivity determined in the supernatant was quite low starting from 1.40 ± 1.27% at 1 h and slightly increasing to 7.27 ± 1.00% after 24 h. The results obtained by us with [^223^Ra]BaFe–CEPA–trastuzumab are comparable with those for gold nanoparticles conjugated with trastuzumab and labeled with ^177^Lu [57] and ^111^In [58] for SK-BR-3 cell line. For example, in the case of trastuzumab–AuNP–^177^Lu, the highest percentage of internalized radioactivity was 76 ± 2% after 5 h [57], which is in very good agreement with our results for [^223^Ra]BaFe–CEPA–trastuzumab with the highest internalization equal to 94.50 ± 0.20% at 6 h of incubation.

The obtained internalization results in radiometric assay were verified by measurements with confocal imaging. The SKOV-3 cells were exposed to bare BaFeNPs, BaFe–CEPA and BaFe–CEPA–trastuzumab for 24 h. Cells treated and untreated with trastuzumab alone were used as reference.

As shown in Figure 9, confocal imaging confirmed that BaFe–CEPA–trastuzumab bioconjugate is effectively internalized by SKOV-3 cells, which is consistent with the results obtained by radiometric method. The small dark spots on the light background reflect NPs (Figure 9b/5) and the red fluorescence signals represent trastuzumab (Figure 9c/5). Merged signals (Figure 9d/5) show that bioconjugate penetrated SKOV-3 cells and localized in the perinuclear space as well as inside the nucleus, passing through the nuclear membrane. As expected, free BaFeNPs and BaFe–CEPA nanoparticles were detected in SKOV-3 cells only in a small amount, as the absence of targeting vector lowers the possibility of internalization. The obtained results are comparable with our previous studies with AuNPs–trastuzumab bioconjugate [40], as well as the results obtained by Cai et al. [57].

### 3.6. In Vitro Cytotoxicity Assay

The cytotoxic effect of non-radioactive bare BaFeNPs and BaFe–CEPA–trastuzumab bioconjugate was tested on SKOV-3 cells by their initial treatment with various concentrations (0.78–400.00 µg/mL) of compounds for 2 h followed by medium removal, supplementation of cells with fresh medium and further incubation for 24, 48, 72 and 96 h. The metabolic activity of the cells was assessed with the MTS assay (Figure 10). The obtained results of cytotoxicity studies for non-radioactive compounds show that BaFeNPs and BaFe–CEPA–trastuzumab are not toxic to the SKOV-3 cells, as cell viability only slightly decreased with increasing concentrations of compounds, finally reaching about 80% at the quite high concentration of 200 µg/mL.

The cytotoxic effect of ^223^Ra radionuclide, functionalized [^223^Ra]BaFe–CEPA nanoparticles and [^223^Ra]BaFe–CEPA–trastuzumab radiobioconjugate was also tested on SKOV-3 cell line by their incubation with different activities of radiocompounds (0.1–50.0 kBq/mL) for 2 h, followed by radioactivity removal and, after supplementation with fresh medium, incubation for another 24 and 48 h with further assessment of metabolic activity using the colorimetric MTS assay (Figure 11). For all tested compounds, SKOV-3 cell viability decreased with increasing radioactivity. The incubation with free ^223^Ra induced only slight degree of cytotoxicity, as the viability of cells slightly decreased with increasing the dose. The surviving cell fraction at 50 kBq/mL for ^223^Ra was still 63.51 ± 3.81% after 24 h and 75.23 ± 7.42% after 48 h. At the same dose, cell viability was adequately 64.97 ± 1.17% at 24 h and 40.68 ± 6.32% at 48 h for [^223^Ra]BaFe–CEPA and 45.64 ± 6.98% and 21.00 ± 1.28% for [^223^Ra]BaFe–CEPA–trastuzumab, respectively.

The calculated IC_50_ values for [^223^Ra]BaFe–CEPA–trastuzumab were 4.96 ± 1.16 kBq/mL at 24 h and 1.39 ± 0.90 kBq/mL at 48 h. It was not possible to determine the IC_50_ values for alone ^223^Ra at both incubation times as well as for [^223^Ra]BaFe–CEPA at 24 h. The IC_50_ for [^223^Ra]BaFe–CEPA at 48 h was 11.29 ± 1.44 kBq/mL, which is about eight-times higher than the value estimated for [^223^Ra]BaFe–CEPA–trastuzumab. The presented results clearly indicate that attachment of trastuzumab molecules to [^223^Ra]BaFe–CEPA nanoparticles increases cytotoxicity effect compared to bare nanoparticles, highlighting that the obtained cytotoxic action of our radiobioconjugate is indeed HER2-receptor-mediated. Similar results were obtained in studies of the cytotoxic effect on spheroids (see Section 3.7). In our previous studies with [^225^Ac]Fe_3_O_4_–CEPA–trastuzumab, we determined the IC_50_ on SKOV-3 cells wasequal to 7.9 ± 1.5 kBq/mL at 48 h [39], which is a much higher value in comparison to the [^223^Ra]BaFe–CEPA–trastuzumab. Considering that the energies deposited by alpha particles emitted by ^223^Ra and ^225^Ac are similar (ca. 28 MeV), the difference in IC_50_ values probably results from the size, shape and degree of dispersion of both radiobioconjugates. The results obtained by us can also be compared with those of Dziawer et al. [40] who conjugated trastuzumab to Au nanoparticles labeled with ^211^At. We observed similar cytotoxicity when considering the differences in the half-life of ^211^At and ^223^Ra and the amount and energy of α particles emitted by ^211^At and ^223^Ra and their decay products.

### 3.7. Radiotoxicity Studies on Cell Spheroids

Microscope images of spheroids treated with 0.78 kBq/mL of ^223^Ra and [^223^Ra]BaFe–CEPA–trastuzumab radiobioconjugate in comparison with control (untreated cells) are shown in Figure 12. Images were made for spheroids on the day of treatment (*D*_0_), and further on Days 4, 8 and 12 after treatment. Spheroid response to the 2-h exposure of various radioactivities of ^223^Ra and [^223^Ra]BaFe–CEPA–trastuzumab is presented in Figure 13.

The results presented as microscope images (Figure 12) and time dependence spheroids growth (Figure 13) indicate that [^223^Ra]BaFe–CEPA–trastuzumab radiobioconjugate significantly inhibits the spheroids growth, regardless of their initial size. The average diameter (ca. 250 µm) of spheroids decreased with time in a dose-dependent manner for [^223^Ra]BaFe–CEPA–trastuzumab, indicating that it is HER2-receptor-mediated, as observed previously in MTS assay. Contrary to the radiobioconjugate, the same doses of ^223^Ra have little to no influence on spheroid growth, as the size of spheroids treated with ^223^Ra and the untreated control is comparable.

The obtained results are consistent with data published by Ballangrud et al. [59], who treated BT-474 (HER2-positive cell line) spheroids with ^225^Ac-labeled trastuzumab. For non-specific antibody labeled with ^225^Ac (control), a significant decrease in spheroid size was noted only for higher doses of 9.25 and 18.5 kBq/mL. This corresponds with our results for ^223^Ra only where SKOV-3 spheroids size slightly but steadily decreased at 12.5 and 50.0 kBq/mL. In the case of ^225^Ac-labeled trastuzumab, a reduction of the spheroid size was observed for 1.85, 3.7 and 18.5 kBq/mL, almost to the same degree regardless of the used dose [59]. We observed a significant decrease in spheroid diameter already after treatment with 0.78 kBq/mL of [^223^Ra]BaFe–CEPA–trastuzumab; however, in our studies, another HER2-positive (SKOV-3) cell line was used. Summarizing, we demonstrated the effectiveness of [^223^Ra]BaFe–CEPA–trastuzumab against spheroids made of clusters of SKOV-3 cells.

## 4. Conclusions

We showed that hydrothermal replacement of Ba^2+^ cations by Ra^2+^ in barium ferrite nanoparticles is a good method for the stable immobilization of the α-emitter ^223^Ra. The demonstrated high receptor affinity and fast internalization of the radiobioconjugate should allow the retention of ^223^Ra and decay products in the targeted site. The high cytotoxicity on SKOV-3 cell monolayers and spheroids indicates that [^223^Ra]BaFe–CEPA–trastuzumab radiobioconjugate may be a potent therapeutic agent against cancerous cells exhibiting overexpression of HER2 receptors, including breast and ovarian tumors. However, due to the short range of α particles, this form of radiotherapy seems to be most appropriate for destroying small tumors and small volume disseminated diseases, rather than larger solid tumors. In addition, the confirmed magnetic properties of the radiobioconjugate should allow for magnet-guided delivery of [^223^Ra]BaFe–CEPA–trastuzumab under the influence of external magnetic field.

## Figures and Tables

**Figure 1 nanomaterials-10-02067-f001:**
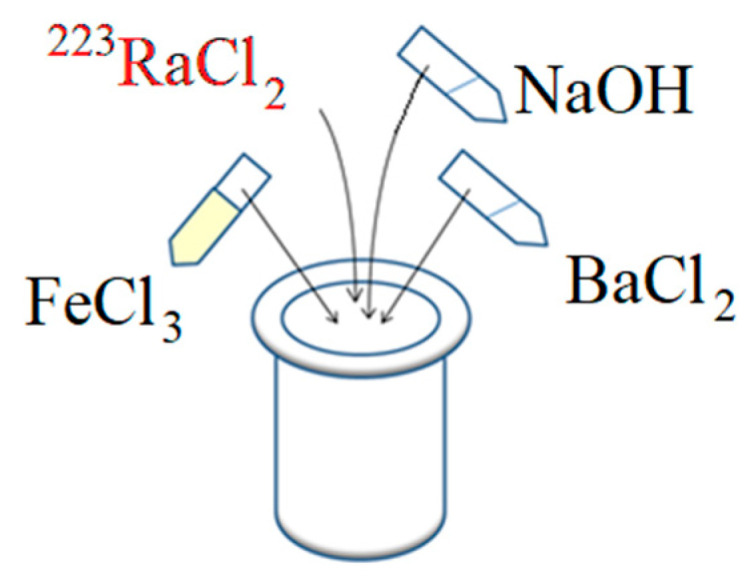
Hydrothermal synthesis of BaFeNPs ([FeCl_3_:BaCl_2_] ratio equal to [8:1], 2.6 M NaOH, 0.1–0.5 MBq ^223^Ra). The temperature of synthesis was 210 °C, with 5 h of continuous stirring.

**Figure 2 nanomaterials-10-02067-f002:**
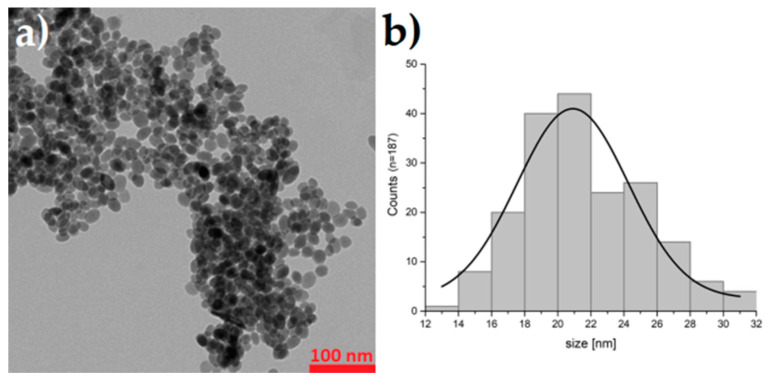
TEM image of synthesized BaFeNPs (**a**); and histogram of nanoparticles sized from TEM image (**b**).

**Figure 3 nanomaterials-10-02067-f003:**
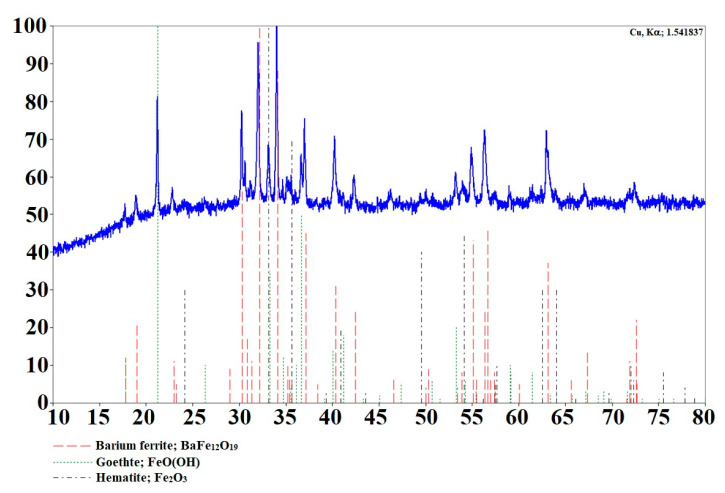
XRD patterns of barium ferrite synthesized hydrothermally at 210 °C.

**Figure 4 nanomaterials-10-02067-f004:**
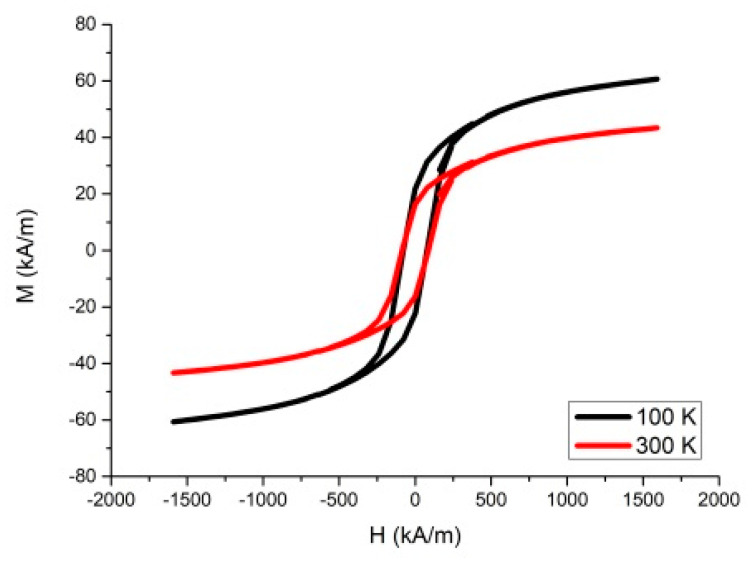
The hysteresis loop of the sample synthesized at 210 °C.

**Figure 5 nanomaterials-10-02067-f005:**
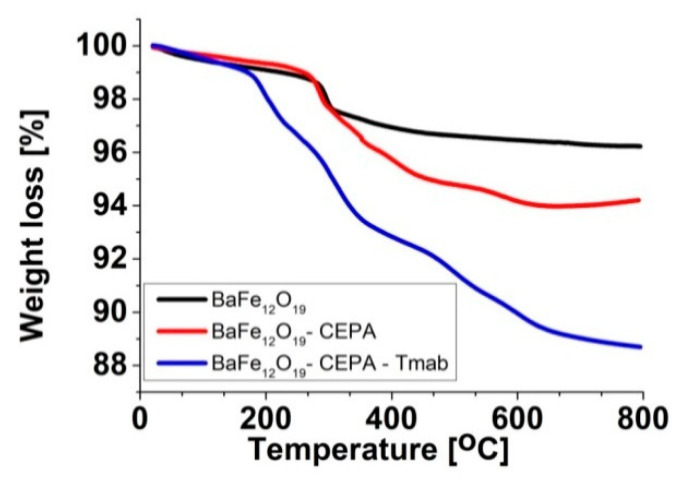
Thermograms of BaFeNPs, BaFe–CEPA NPs and BaFe–CEPA–trastuzumab bioconjugate.

**Figure 6 nanomaterials-10-02067-f006:**
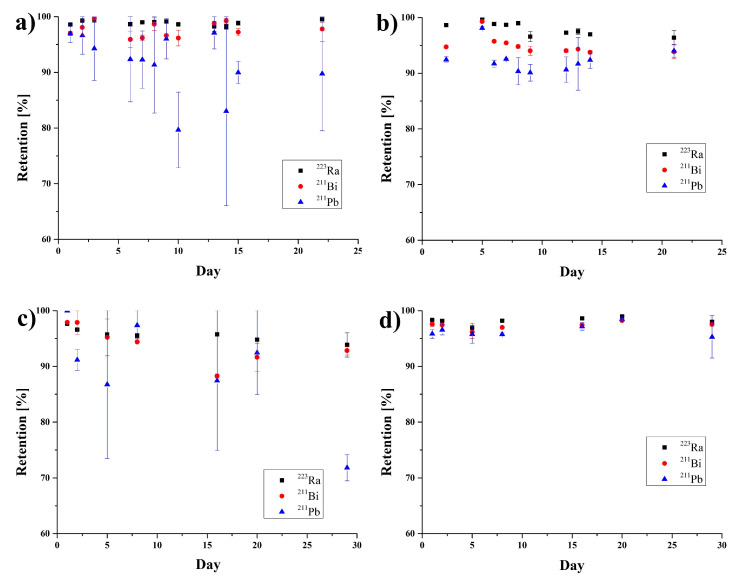
Stability studies of [^223^Ra]BaFeNPs in (**a**) 0.9% NaCl and (**b**) human serum; and [^223^Ra]BaFe–CEPA–trastuzumab bioconjugate in (**c**) 0.9% NaCl and (**d**) and human serum.

**Figure 7 nanomaterials-10-02067-f007:**
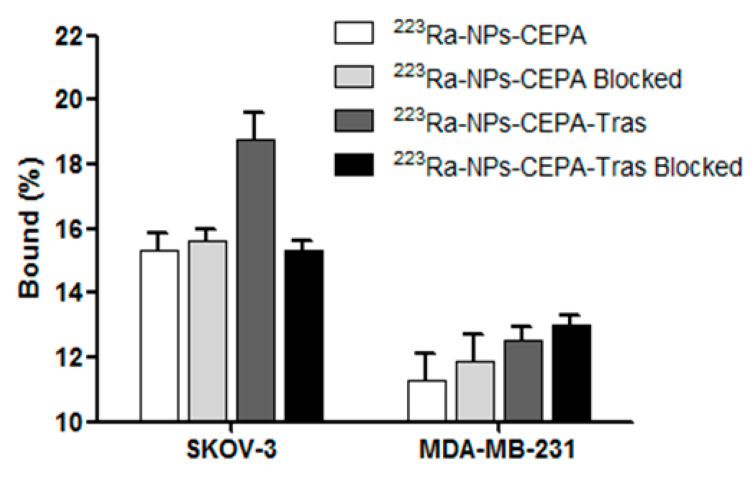
Specificity of binding on SKOV-3 (HER2+) and MDA-MB-231 (HER2-) cells. [^223^Ra]BaFe–CEPA and [^223^Ra]BaFe–CEPA–trastuzumab radiobioconjugate were incubated in the presence (blocked) or absence of a 100-fold molar excess of trastuzumab.

**Figure 8 nanomaterials-10-02067-f008:**
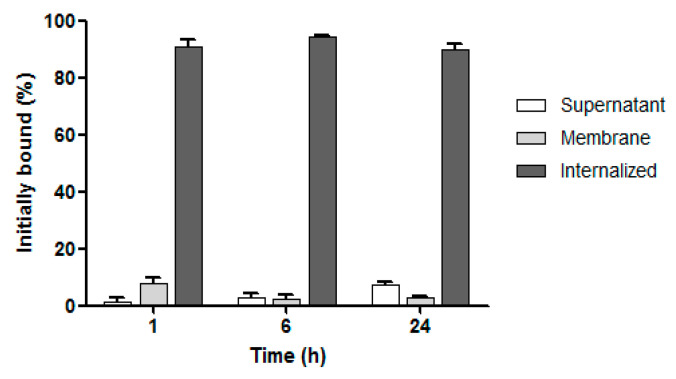
Internalization of [^223^Ra]BaFe–CEPA–trastuzumab radiobioconjugate in SKOV-3 cells.

**Figure 9 nanomaterials-10-02067-f009:**
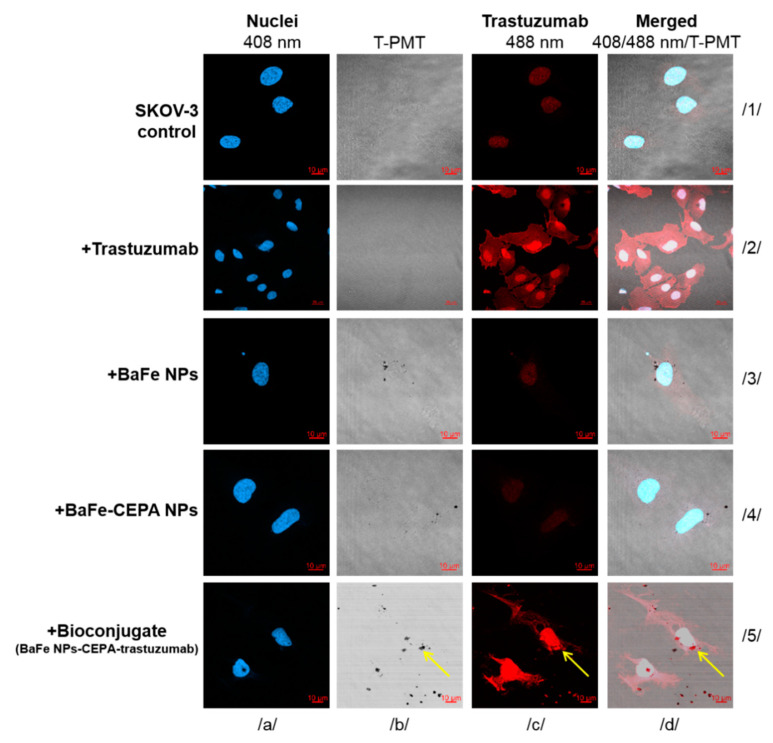
Internalization of the bare BaFeNPs, BaFe–CEPA and BaFe–CEPA–trastuzumab bioconjugate by SKOV-3 cells determined by confocal microscopy. Cells treated and untreated with trastuzumab are shown as reference. Fluorescence signals present: red, trastuzumab distribution; blue, nuclei localization; dark spots, nanoparticles visualized with a transmitted light detector (T-PMT). The merged images are presented in the last column. The yellow arrows show the subcellular localization of the BaFe–CEPA–trastuzumab bioconjugate. The letters a–d and the numbers 1–5 have been included to locate the individual images.

**Figure 10 nanomaterials-10-02067-f010:**
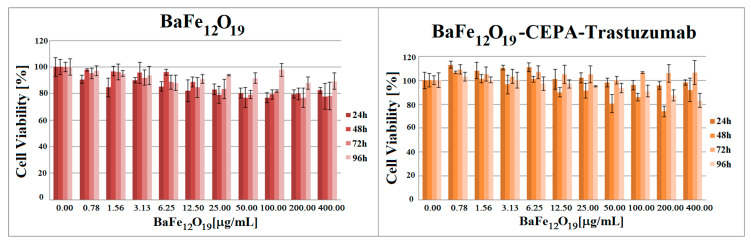
Viability of SKOV-3 cells after treatment with different concentrations of bare BaFeNPs and BaFe–CEPA–trastuzumab bioconjugate. Cells were incubated for 24, 48, 72 and 96 h, after which their viability was measured using MTS assay. The results are expressed as a percentage of control cells.

**Figure 11 nanomaterials-10-02067-f011:**
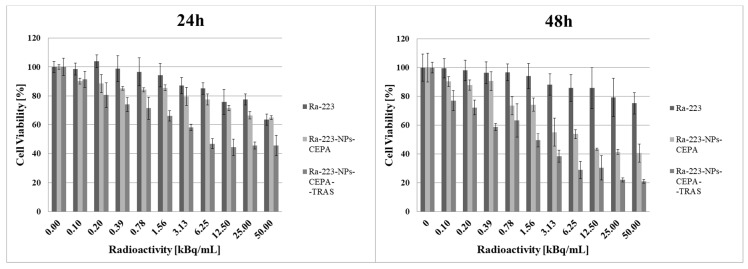
Viability of SKOV-3 cells after treatment with different radioactivity of free ^223^Ra radionuclide, functionalized [^223^Ra]BaFe–CEPA nanoparticles and [^223^Ra]BaFe–CEPA–trastuzumab radiobioconjugate. The results are expressed as a percentage of control cells.

**Figure 12 nanomaterials-10-02067-f012:**
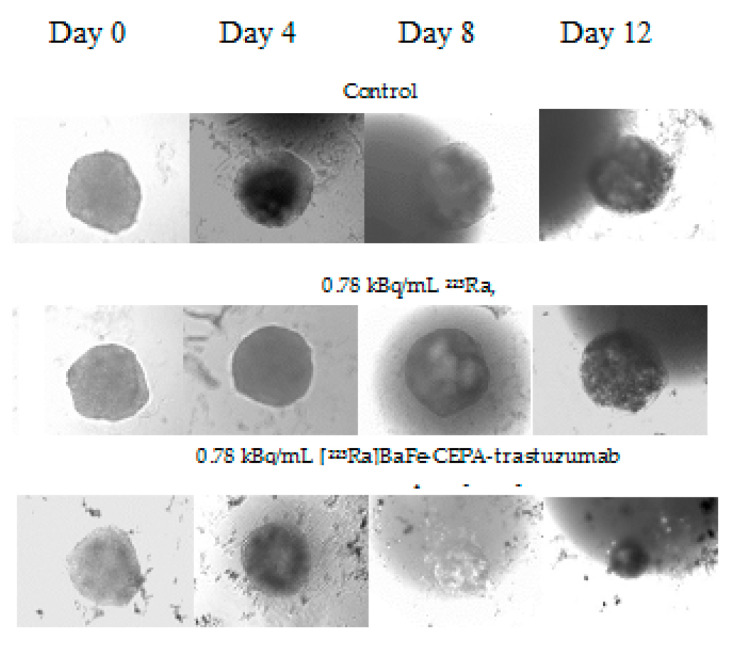
Microscope images of representative spheroids. Images made for 250-µm spheroids on the day of treatment (*D*_0_) and on Days 4, 8 and 12 after treatment.

**Figure 13 nanomaterials-10-02067-f013:**
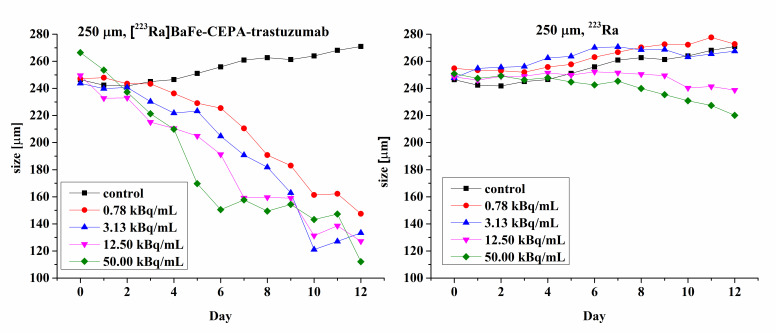
Time dependence spheroids growth.

**Table 1 nanomaterials-10-02067-t001:** The hydrodynamic diameters and zeta (ζ) potentials of bare BaFeNPs, BaFe–CEPA and BaFe–CEPA–trastuzumab in water determined by DLS method.

	BaFeNPs	BaFe–CEPA	BaFe–CEPA–Trastuzumab
Hydrodynamic diameter (nm)	144.7 ± 6.6	59.4 ± 2.4	99.9 ± 3.0
Zeta potential (mV)	+26.4 ± 1.6	−23.1 ± 1.3	+27.2 ± 0.7

**Table 2 nanomaterials-10-02067-t002:** The hydrodynamic diameters and zeta (ζ) potentials of bare BaFeNPs, BaFe–CEPA and BaFe–CEPA–trastuzumab in 1 mM PBS determined by DLS method.

	BaFeNPs	BaFe–CEPA	BaFe–CEPA–Trastuzumab
Hydrodynamic diameter (nm)	116.6 ± 6.0	60.9 ± 6.6	218.3 ± 3.7
Zeta potential (mV)	−40.0 ± 1.1	−49.6 ± 0.9	−8.8 ± 0.7

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
