# Peer review of "Trastuzumab Modified Barium Ferrite Magnetic Nanoparticles Labeled with Radium-223: A New Potential Radiobioconjugate for Alpha Radioimmunotherapy"

_nanomaterials, 2020, doi:10.3390/nano10102067_

Round 1

Reviewer 1 Report

The manuscript describes the preparation and characterization of novel Ra-223  Barium ferrite NPs modified by trastuzumab antibody together with both positive and negative in vitro test controls. 

I have just few suggestions and questions:   

Please describe in more detail the stability tests. Particularly focus on the methodology of recoil atoms measurement in the supernatant. Were the tests performed under static conditions? Were the samples shaken/mixed? Did you observe some radionuclide sorption on the surface of vials? Resorption of daughters on the surface of NPs may give you false-positive results specially when the time intervals are in the order of days. Here some addition of competing chelator to the solution should give you an idea on true radionuclide release. Please describe the method shortly in the text.  

Table 1. - It is quite interesting comparison between PBS and water dispersions. It seems that contrary to the conjugate, the raw NPs are more stable in PBs. Could you provide DLS / Z-potential measurement of the conjugate in plasma, cell medium or some simulated biological fluid, that should give an idea about real hydrodynamic dimensions in vitro/ in vivo? 

Chapter 3.4. - Could you please compare your results with some additional literature refering to nanoconstructs or small bioconjugates decorated with HER(+) targeting moieties? What was the ratio between the cell number and the applied NPs? This may significantly impact your specific binding study. 

Page 15, line 585 - please explain how did the free Ra-223 appeared in the test? What effect was observed? Wasn´t the IC50 reached? Please refer to the spheroids experiment here.  

Figure 11 - Please correct the units of radioactivity concentration to kBq/ml. 

Author Response

1. Please describe in more detail the stability tests. Particularly focus on the methodology of recoil atoms measurement in the supernatant. Were the tests performed under static conditions? Were the samples shaken/mixed? Did you observe some radionuclide sorption on the surface of vials? Resorption of daughters on the surface of NPs may give you false-positive results specially when the time intervals are in the order of days. Here some addition of competing chelator to the solution should give you an idea on true radionuclide release. Please describe the method shortly in the text.  

The process was carried out under static conditions. Details of the experiment have been added to the experimental section. We agree with the reviewer that experiments under static conditions cannot be transferred to in vivo models, as the blood flow may rapidly dislocate the decay products from the surface of the nanoparticles and reduce the readsorption probability. This was indicated in the text (line 456-460).

2. Table 1. - It is quite interesting comparison between PBS and water dispersions. It seems that contrary to the conjugate, the raw NPs are more stable in PBs. Could you provide DLS / Z-potential measurement of the conjugate in plasma, cell medium or some simulated biological fluid, that should give an idea about real hydrodynamic dimensions in vitro/ in vivo? 

Unfortunately, our DLS studies in serum have not produced reasonable results. Proteins present in the serum interfered with the measurement.

3. Chapter 3.4. - Could you please compare your results with some additional literature refering to nanoconstructs or small bioconjugates decorated with HER(+) targeting moieties? What was the ratio between the cell number and the applied NPs? This may significantly impact your specific binding study. 

Unfortunately, it is very difficult to compare the receptor affinity, internalization and cytotoxicity of different trastuzumab-containing bioconjugates. The results depend on the type of nanoparticle, its size, shape, linker used, amount of trastuzumab and the type of radionuclide. Affinity for HER2 receptors, high internalization as well as a perinuclear localization are observed in all the studies. The results obtained by us can be compared with the results in the publication of Dziawer et al. Nanomaterials 2019, 9, 632 where 4 trastuzumab particles were attached to 5 nm gold nanoparticles and alpha emitter 211At was attached. We observe very similar internalization and receptor affinity of the bioconjugates. Interestingly, we also observe similar cytotoxicity when taking into account the differences in the half-life of 211At and 223Ra and the amount of particles emitted by 211At and 223Ra and its decay products. The relevant sentences were added to the manuscript. line 507-511 and 566-576

4. Page 15, line 585 - please explain how did the free Ra-223 appeared in the test? What effect was observed? Wasn´t the IC50 reached? Please refer to the spheroids experiment here.  

In both cellular and spheroidal experiments we added for comparison free 223Ra.  However, the cytotoxicity was much lower and we did not obtain in our radioactivity range a 50% cell  viability. A similar tendency was observed for the spheroids. We have added the appropriate sentence to the manuscript. 

5. Figure 11 - Please correct the units of radioactivity concentration to kBq/ml.

done

Reviewer 2 Report

Few comments regarding the methodological approach and presentation of results:

  1. The data on the stability of trastuzumab after binding to BaFeNPs are not available, thus there is no possibility to evaluate the suitability of binding procedure and functionality of NPs.
  2. The application of a single spectrophotometric determination method for evaluation of trastuzumab concentration has to be confirmed considering the stability/functionality of MAB. (line 208)
  3. The possible effect of drying step on the performance of MAB for binding HER-2 should be considered. (line 272)
  4. The arguments supporting statements on possible causes of observed instabilities of NPs (lines 443 and 446) are recommended to be provided.
  5. High physical sorption of NPs (line 513) question the functionality of trastuzumab binding to NPs and it should be confirmed by additional argumentation, particularly considering 3% (only) increased SKOV-3 binding.
  6. Arguments for not presenting the internalization data for MDA-MB-231 cells should be provided.
  7. The wording "non-toxic effect" (line 566) should be reconsidered.  

Author Response

  1. The data on the stability of trastuzumab after binding to BaFeNPs are not available, thus there is no possibility to evaluate the suitability of binding procedure and functionality of NPs.

As mentioned in the manuscript, procedure of trastuzumab conjugation was very simple "After activation of carboxylic groups in CEPA to NHS-ester, the nanoparticles were separated from the reaction mixture on the magnet, washed a few times with 10 mM PBS and resuspended in 1 mL of 10 mM PBS, pH 7.4. Then, 250 μg (1.7 nmoles) of trastuzumab was added to the suspended nanoparticles. The synthesis of bioconjugate with trastuzumab was performed overnight. The conjugation product was separated from the solution using the solid magnet, washed with distilled water and used for further studies". This procedure is very similar to the procedure of chelator attachment to trastuzumab, where no antibody damage is observed. In our studies, after attachment, trastuzumab retains its properties, as shown by the results of cellular tests.

2. The application of a single spectrophotometric determination method for evaluation of trastuzumab concentration has to be confirmed considering the stability/functionality of MAB. (line 208)

In our experiments, we used trastuzumab isolated from the drug Herceptin, where the activity is repeatedly confirmed. Also our studies of free trastuzumab on spheroids confirm its activity. We observed the characteristic internalization of trastuzumab and its accumulation in perinuclear space (Fig.9)

3. The possible effect of drying step on the performance of MAB for binding HER-2 should be considered. (line 272)

The BaFeNPs, BaFe-CEPA and BaFe-CEPA-trastuzumab bioconjugate were dried at RT only for thermogravimetric experiments where we determined the amount of trastuzumab in bioconjugates. In other experiments, we used bioconjugates in solution. They were not dried at any stage of the synthesis and were magnetically insolated.

4. The arguments supporting statements on possible causes of observed instabilities of NPs (lines 443 and 446) are recommended to be provided.

Thermogravimetry is commonly used in nanotechnology to determine the amount of biomolecules attached to the surface of a nanoparticle. The decomposition of organic substances at high temperatures is a source of weight loss. From the loss of mass, we determine the number of particles attached to the nanoparticle. This is a good method as has been confirmed in our paper by radiometric method using radioactive [131I]-trastuzumab. Aditional sentence was added. 

5. High physical sorption of NPs (line 513) question the functionality of trastuzumab binding to NPs and it should be confirmed by additional argumentation, particularly considering 3% (only) increased SKOV-3 binding.

Due to sedimentation of nanobioconjugates during long incubation time, in our experiments the radiobioconjugates [223Ra] BaFe-CEPA and [223Ra] BaFe-CEPA-trastuzumab were incubated at 4 oC for only 2 h (line 294-295). Physical adsorption in short time plays a smaller role. Physical adsorption is evidenced by binding on cells previously blocked with trastuzumab. In future experiments, we will try to reduce this effect by using the cell culture under flow instrument. Few sentence was added to the text. 

6. Arguments for not presenting the internalization data for MDA-MB-231 cells should be provided.

Due to the lack of specific binding, we did not conduct  internalization studies on MDA-MB-231 cells. The sentence was added to the text. 

7. The wording "non-toxic effect" (line 566) should be reconsidered.  

done

Reviewer 3 Report

The authors report about modified barium ferrite nanoparticles for radioimmunotherapy and magnetic hyperthermia. The text is clearly written and can be followed with no major issues.

Comments about the document are given below:

Introduction: although I understand the authors want to present an overview of the topic, it seems to me to be quite long. I would suggest the authors to summarize it a little (perhaps using a table) since the work motivation should be clearly stated and fully understood by the audience.  

Experimental: Proper description presented.

Results:

- crystallinities: I believe you mean crystallite.

- the coercivity of magnetic materials is strongly dependent on T. However, based on Fig. 4, it is constant for a T window of 200K! Why is this happening?

- why do the authors indicate the "soft magnetic properties" of the material under evaluation? The Hc values obtained will characterize it as a hard magnetic material. Please explain.

- please indicate in Fig. 5 the source of each mass loss so that readers can have the full information onto a single image.

- please change the scales of Fig. 6 from 0-100 to maybe 70-100. 

- why on Fig. 10 solid bars are also above 100%? I understand that error bards could surpass such value, but not the solid ones.

- a missing point of the work refers to the magnetic hyperthermia aspect. No data on heating/cooling curves are presented, as well as SAR info. Therefore, the authors must include content on the subject or modify the title of the manuscript.

Author Response

1. Introduction: although I understand the authors want to present an overview of the topic, it seems to me to be quite long. I would suggest the authors to summarize it a little (perhaps using a table) since the work motivation should be clearly stated and fully understood by the audience. 

The introduction has been significantly shortened.

2. crystallinities: I believe you mean crystallite.

done, thank you

3. the coercivity of magnetic materials is strongly dependent on T. However, based on Fig. 4, it is constant for a T window of 200K! Why is this happening?

However, as can be seen in Fig. 4, the coercivity in 100 K is significantly greater than in 300 K. We cannot explain why this difference is not large.

4. why do the authors indicate the "soft magnetic properties" of the material under evaluation? The Hc values obtained will characterize it as a hard magnetic material. Please explain.

This is obviously a mistake. We improved the text.

5. please indicate in Fig. 5 the source of each mass loss so that readers can have the full information onto a single image.

We decided to describe the curves under the figure. We belive that it would be easier to read.

6. please change the scales of Fig. 6 from 0-100 to maybe 70-100. 

Done. Very good advice.

7. why on Fig. 10 solid bars are also above 100%? I understand that error bards could surpass such value, but not the solid ones.

Often, in cell studies, small doses of radioactivity or the presence of nanoparticles can increase cell viability.

8. a missing point of the work refers to the magnetic hyperthermia aspect. No data on heating/cooling curves are presented, as well as SAR info. Therefore, the authors must include content on the subject or modify the title of the manuscript.

The initial goal of our work was to use 223 Ra-labeled BaFe for hyperthermia and radiotherapy. Unfortunately, the low saturation magnetization of hydrothermally synthesized BaFeNPs significantly reduce their applicability in magnetic hyperthermia, where high saturation magnetization of the samples is needed. We agree with the reviewer and we changed the title of the publication.

Reviewer 4 Report

Reviewers' comments:

(a) ARTICLE RANKING

* Above average but may be further improved

(b) RECOMMENDATION

* Minor revision. With minor revisions I expect the article to make a significant contribution to the literature. 

(c) COMMENTS of Reviewer

* Fairly good paper describing a useful method and material. It contains, however, several technical and formal flaws so that a major revision is mandatory. If these are eliminated, the paper will be much more attractive. In more detail:

TITLE:
* Title: Remove 223Ra, Remember that researchers usually do not electronically search for such terms. Mechanistic description should be given in the abstract.

ABSTRACT:
* The few sentence is presenting trivial information and is not a result of the work presented here. Readers of journal are experts. Also note that most methods based on the use of nanomaterials are not simple but require substantial skills in order to be reproduced.

* Specify the zeta potential observed, the response to pH and the biocompatibility range. 

INTRODUCTION
* Introduction: The introduction section also contains trivial statements. This section lack in connectivity and citation of recent literature. Otherwise readers will be bored when reading this. The introduction should be reduced in length and have a focus on current literature.

EXPERIMENTAL PART

* The part “chemical reagents, radionuclides and cell lines is a mixing. This latter information should be presented separately. Also, please provide the instrumentation for measuring zeta potential.

* Altogether, the experimental part is too long. A considerable part can be shifted to the Supplementary Material.

RESULTS
* Synthesis and characterization. The some paragraph duplicates information provided in the Experimental Part. It should be reduced in length.

* Some experimental data Figure 13 lack (relative) standard deviations. Give averaged data for important experimental data along with standard deviations (±) and the number of experiments (n = ?). In case of particle sizes, specify the number of particles that has been taken into account to calculate data. Give error bars in figures.

* I was surprised to see that the method, according to the manuscript (discussion & conclusion), has no major limitations. Is this correct? If not, please specify at the end of the section

CONCLUSIONS
* Do not duplicate text of the Abstract. Focus on conclusions, on the scope (such as applicability to related detection schemes), and also on limitations, but do not repeat figures of merit. 

Author Response

Remove 223Ra, Remember that researchers usually do not electronically search for such terms. Mechanistic description should be given in the abstract.

Done

The few sentence is presenting trivial information and is not a result of the work presented here. Readers of journal are experts. Also note that most methods based on the use of nanomaterials are not simple but require substantial skills in order to be reproduced.

As suggested by the reviewer, we shortened the abstract by a few unnecessary sentences.

Specify the zeta potential observed, the response to pH and the biocompatibility range. 

done

Introduction: The introduction section also contains trivial statements. This section lack in connectivity and citation of recent literature. Otherwise readers will be bored when reading this. The introduction should be reduced in length and have a focus on current literature.

In the introduction, we decided to show the reader what solutions have been used to immobilize radium radionuclides. To our knowledge, we cited all the papers published so far, including those from 2020. In agreement with the reviewer, we shortened the introduction considerably.

The part “chemical reagents, radionuclides and cell lines is a mixing. This latter information should be presented separately. Also, please provide the instrumentation for measuring zeta potential.

We separated points chemical reagents, radionuclides and cell lines. As mentioned in the manuscript the hydrodynamic diameter and zeta potential were determined with dynamic light scattering (Zetasizer Nano ZS DLS, Malvern, UK)

Altogether, the experimental part is too long. A considerable part can be shifted to the Supplementary Material.

It is difficult to decide what part to move to the Supplementary Material and what to leave in the experimental, so we decided to leave it in the text, slightly shortening it.

RESULTS
Synthesis and characterization. The some paragraph duplicates information provided in the Experimental Part. It should be reduced in length.

According to the reviewer's remark, we moved the description of synthesis and figure to experimental part.

Some experimental data Figure 13 lack (relative) standard deviations. Give averaged data for important experimental data along with standard deviations (±) and the number of experiments (n = ?). In case of particle sizes, specify the number of particles that has been taken into account to calculate data. Give error bars in figures.

As the cultured spheroids differ in size (+/-10 µm) and shape (as can be seen on figure in the Day 0), we have presented the graph for one selected spheroid.

I was surprised to see that the method, according to the manuscript (discussion & conclusion), has no major limitations. Is this correct? If not, please specify at the end of the section

We believe this is correct

CONCLUSIONS
* Do not duplicate text of the Abstract. Focus on conclusions, on the scope (such as applicability to related detection schemes), and also on limitations, but do not repeat figures of merit. 

Conclusions have been shortened as suggested by the reviewer.